# The Importance of Molecular Genetic Testing for Precision Diagnostics, Management, and Genetic Counseling in MODY Patients

**DOI:** 10.3390/ijms25126318

**Published:** 2024-06-07

**Authors:** Lăcrămioara Ionela Butnariu, Delia Andreia Bizim, Carmen Oltean, Cristina Rusu, Monica Cristina Pânzaru, Gabriela Păduraru, Nicoleta Gimiga, Gabriela Ghiga, Ștefana Maria Moisă, Elena Țarcă, Iuliana Magdalena Starcea, Setalia Popa, Laura Mihaela Trandafir

**Affiliations:** 1Department of Medical Genetics, Faculty of Medicine, “Grigore T. Popa” University of Medicine and Pharmacy, 700115 Iasi, Romania; abcrusu@gmail.com (C.R.); setalia_popa@yahoo.com (S.P.); 2Department of Diabetes, Saint Mary’s Emergency Children Hospital, 700309 Iasi, Romania; delia_biz@yahoo.com (D.A.B.); oltean_carmen@yahoo.com (C.O.); 3Department of Mother and Child, Faculty of Medicine, “Grigore T. Popa” University of Medicine and Pharmacy, 700115 Iasi, Romania; paduraru.gabriela@umfiasi.ro (G.P.); nicoleta.chiticariu@umfiasi.ro (N.G.); gabriela.ghiga@umfiasi.ro (G.G.); stefana-maria.moisa@umfiasi.ro (Ș.M.M.); magdalenastarcea@gmail.com (I.M.S.); laura.trandafir@umfiasi.ro (L.M.T.); 4Department of Surgery II—Pediatric Surgery, “Grigore T. Popa” University of Medicine and Pharmacy, 700115 Iasi, Romania; elena.tuluc@umfiasi.ro

**Keywords:** monogenic diabetes mellitus, MODY, genetic variant, gene panel, genetic counseling

## Abstract

Maturity-onset diabetes of the young (MODY) is part of the heterogeneous group of monogenic diabetes (MD) characterized by the non-immune dysfunction of pancreatic β-cells. The diagnosis of MODY still remains a challenge for clinicians, with many cases being misdiagnosed as type 1 or type 2 diabetes mellitus (T1DM/T2DM), and over 80% of cases remaining undiagnosed. With the introduction of modern technologies, important progress has been made in deciphering the molecular mechanisms and heterogeneous etiology of MD, including MODY. The aim of our study was to identify genetic variants associated with MODY in a group of patients with early-onset diabetes/prediabetes in whom a form of MD was clinically suspected. Genetic testing, based on next-generation sequencing (NGS) technology, was carried out either in a targeted manner, using gene panels for monogenic diabetes, or by analyzing the entire exome (whole-exome sequencing). *GKC*-MODY 2 was the most frequently detected variant, but rare forms of *KCNJ11*-MODY 13, specifically, *HNF4A*-MODY 1, were also identified. We have emphasized the importance of genetic testing for early diagnosis, MODY subtype differentiation, and genetic counseling. We presented the genotype–phenotype correlations, especially related to the clinical evolution and personalized therapy, also emphasizing the particularities of each patient in the family context.

## 1. Introduction

Maturity-onset diabetes of the young (MODY) is part of the heterogeneous group of monogenic diabetes (MD) characterized by pancreatic β-cell dysfunction, which also includes neonatal diabetes (ND), maternally inherited diabetes with deafness and diabetes-associated syndromes (such as the Wolfram, Bardet–Biedl, Wolcott–Rallison, and Mitchell–Riley syndromes) [1,2,3].

According to the criteria reported in 2008, MODY is defined by: early-onset non-insulin-dependent diabetes mellitus (NIDDM) (usually <25 years); an autosomal dominant inheritance (there is at least one other affected family member, most often a multigenerational transmission); the non-immune dysfunction of pancreatic β-cells (the absence of autoantibodies) associated with primary insulin secretion defects; and the absence of metabolic syndrome [4,5,6].

The prevalence of MODY among pediatric patients, adolescents, and young people is constantly increasing. MODY cases represent approximately 1–6% of all cases of diabetes mellitus (DM) [1,2,3]. The diagnosis of MODY still remains a challenge for clinicians, with many cases being misdiagnosed as type 1 or type 2 diabetes mellitus (T1DM/T2DM), and over 80% of cases remaining undiagnosed [7].

The phenotypic overlaps of MODY with those of common forms of diabetes can cause diagnostic errors. Patients with MODY can be underweight young people, as in the case of T1DM, but do not require insulin, and antibodies against pancreatic beta-cells are absent, or they can have an appearance similar to that of T2DM, with obesity. Family history of diabetes can be positive in both T1DM, and T2DM, as well as in the case of MODY. Moreover, in some cases of MODY, family history of diabetes is negative, the cause being either a de novo mutation or the lack of accuracy of information regarding diabetes in the patient’s parents and other relatives. A correct diagnosis of MODY is important for the establishment of treatment, as appropriate treatment is determined by the etiology of the disease (e.g., oral sulfonylurea treatment of *HNF1A* and *HNF4A* MODY compared to insulin in T1DM), but can only be confirmed by molecular genetics testing. Therefore, misdiagnosis can lead to inadequate treatment. In the context of the increase in the general prevalence of obesity, the clinical differential diagnosis between MODY and T2DM can be difficult, which underlines the importance of genetic testing that allows the correct diagnosis. Moreover, unequal access to genetic analysis could be a contributing factor to the delayed diagnosis and misdiagnosis in the case of patients with MODY [6,7].

With the introduction of modern technologies such as next-generation sequencing, (NGS), using targeted gene panels or whole-exome sequencing (WES) and whole-genome sequencing (WGS) after 1990, important steps were taken to decipher the complex molecular mechanisms and heterogeneous etiology of monogenic diabetes (MD), including MODY (Figure 1).

Thus, 14 involved genes and loci were identified, which led to a reclassification of MODY according to the genetic etiology, in all cases, with the mutations being transmitted in an autosomal dominant manner (Table 1) [1,3,6,8,9,10,11].

MODY represents the most common type of monogenic diabetes (MD) in Europe, accounting for about 2–5% of all diabetes cases [11]. Mutations in the *HNF1A* (MODY 3), *GCK* (MODY 2), *HNF4A* (MODY 1), and *HNF1B* (MODY 5) genes cause more than 95% of MODY cases in the Caucasian population. Differences are reported related to the spectrum of variants detected in Europeans compared to Asians [11]. Moreover, differences are reported regarding the prevalence of MODY in different European populations, correlated with the genetic etiology [3,11]. *GCK*-MODY (MODY 2) and *HNF1A*-MODY (MODY 3) account for approximately 30–60% of all MODYs. *GCK*-MODY has a higher prevalence in Italy, France, Germany, Spain, and the United States [3]. The most common form of MODY detected in Great Britain is *HNF1A*-MODY 3 (21–64%), while, in France, *GCK*-MODY 2 was detected in 8 to 63% of cases. In 10% of MODY cases, the *HNF4A* (MODY 1) and *HNF1B* (MODY 5) variants are detected. The other forms of MODY are rare, being described only in a few families [10].

Pathogenic variants in *ABCC8*, which encodes sulfonylurea receptor 1 (SUR1), and *KCNJ11*, which encodes inwardly rectifying potassium channel (Kir6.2), both subunits of the ATP-sensitive potassium channel (K_ATP_) in pancreatic β-cells (Figure 1, Table 1), are common causes of permanent and transient neonatal diabetes (TND/PND). Rarely, mutations of the two genes cause diabetes in children or young adults, classified as *ABCC8*-MODY 12, specifically, *KCNJ11*-MODY 13 [12].

It is considered that many of the MODY cases remain undiagnosed, as no mutations have been identified in the 14 already known genes, which are included in the MODY-X subtype. In their case, the involvement of additional genes and loci, not yet identified, is suspected [11]. In their case, an extended analysis such as whole-exome sequencing (WES) or whole-genome sequencing (WGS) could bring additional information with which to elucidate the etiology [3,9]. It is known that mutations in some genes that cause MODY are also associated with other forms of diabetes (allelic and locus heterogeneity in the case of the *NEUROD1*, *KLF11*, *INS*, *ABCC8*, *KCNJ11*, and *APPL1* genes) (Table 1).

Molecular genetic testing of patients is important because it allows the identification of the mutations, thus confirming the diagnosis of MODY, classifying the subtype, predicting the likely clinical course, and influencing the patient’s treatment.

The first-line treatment in *HNF4A*-MODY 1 and *HNF1A*-MODY 3 is represented by sulfonylureas [13,14]. Most patients with *GKC*-MODY 2 are asymptomatic, and the risk of vascular complications is low; therefore, they do not require pharmacological treatment [11,12,13,14,15].

Genetic counseling is a major step in the management of patients with MODY, taking into account the fact that the risk of recurrence of the disease in first-degree relatives of the proband is 50%, the disease having an autosomal dominant inheritance. Genetic testing in family members of affected individuals allows the pre-symptomatic diagnosis of mutation carriers, for whom preventive measures are required, which include regular blood glucose monitoring, early diagnosis, and appropriate treatment.

The aim of our study was to identify the genetic variants associated with MODY in a group of patients with early-onset diabetes/prediabetes in whom there was clinical suspicion of the monogenic form of diabetes. We also presented the genotype–phenotype correlations, emphasizing the particularities of each patient in the context of his family correlated with the type of mutation detected, especially related to the clinical evolution and personalized therapy.

## 2. Results

Extended genetic testing (gene panel for monogenic diabetes and WES) in the case of the eight patients with MODY revealed that six of them had a variant of *GKC-*MODY 2, and two other patients had rare variants of *KCNJ11*-MODY 13, specifically, *HNF4A*-MODY 1 (Table 2). All eight mutations detected in patients with MODY were pathogenic, six of which were missense, one nonsense mutation, and a heterozygous splice donor variant (Table 1).

In three of the patients with *GCK*-MODY 2, the heterozygous missense variant *GCK* c.106C>T (p.Arg36Trp) located in exon 2 of the *GKC* gene was identified (Figure 2). Another heterozygous missense variant *GCK* c.617C>T (p.Thr206Met) located in exon 6 was detected in one of the patients with MODY, and the nonsense variant *GCK* c.1072C>T (p.Arg358*) located in exon 9 gene was identified in another patient. In the case of one patient, a heterozygous splice donor variant *GCK* c.1019+1G>A was detected (Table 3 and Figure 2). The missense variant *KCNJ11* c.616C>T (p.Arg206Cys) located in exon 1 of the gene is considered a rare variant of MODY, with, in most cases, *KCNJ11* mutations being associated with permanent and transient neonatal diabetes (TND/PND), rarely being detected in children or young adults with diabetes. Another rare missense variant was detected in one of the patients and was located in exon 7 of the *HNF4A* gene c.733C>T (p.Arg245Cys). Other *HNF4A* variants are associated with forms of late-onset non-insulin-dependent diabetes mellitus (late-onset NIDDM) (OMIM 125853) [10].

Clinical and anamnestic data and biochemical parameters of patients with MODY are presented in the Table 3.

The age of the patients at the time of diagnosis varied between 2.5 and 15.9 years; seven of the eight patients were male and there was only one female patient (P08, E.T.). The family history for DM was positive in seven of the eight cases. In the case of patient P08, the family history of DM was negative, the mutation being most likely de novo (Figure 3).

In all patients with prediabetes/diabetes, the plasma level of C-peptide and at least two types of antibodies were initially measured—anti-GAD65 (glutamic acid decarboxylase 65), and one of the following anti-IA2 (tyrosine phosphatase-related islet antigen 2) or anti-ICA (Islet cell antibodies)—in order to exclude immune-mediated type 1 diabetes (T1D) (Table 3).

The HbA1c value even slightly increased with a positive family history, and negative antibodies required genetic testing (the cost of which was supported by the family).

The evolution of biochemical markers (hyperglycemia, serum C-peptide level, HbA1c, anti-GAD65, anti-ICA, and anti-IA2 antibodies) is presented in Table 3. The higher level of glycemia varied between 140 and 248 mg/dL, while those of HbA1c were between 5.9% and 11.5% (Table 3). Two of the eight patients presented clinical symptoms (polyuria, polydipsia, and metabolic syndrome), and one of the cases was associated with ketoacidosis (DKA). Obesity was detected in three of the patients (P06, P07, and P08). Five of the patients (P01, P02, P04, P05, and P06) did not require pharmacological treatment; only one patient (P03) continued insulin initiated before the genetic testing, one patient (P07) benefited from insulin therapy associated with sulfonylurea, and one patient (P08) was treated with sulfonylurea associated with diet (Table 3).

## 3. Discussions

In the majority of patients with MODY included in the study (six of the eight patients), molecular genetic testing (gene panel for MODY or WES) identified the presence of a mutation in the *GKC* gene (MODY 2). In three of the patients (P01, P02, and P03), the same pathogenic variant *GCK* c.106C>T was detected (Table 2, Figure 3). Five of the six patients with *GCK*-MODY 2 (P01, P02, P04, P05, and P06) did not require pharmacological treatment; in their case, only diet and blood glucose monitoring were recommended. In the case of one of the patients (P03), diet was initially recommended, but, due to the increasing of the HbA1c value from 6.2% to 6.7%, the basal insulin therapy initiated before the genetic testing was maintained.

In the case of the first patient (P01, I.C., 2 years old), during a routine control, elevated plasma glucose values (150 mg/dL) associated with HbA1c values between 5.9% and 6.1% were identified in the absence of other clinical symptoms (Table 3). Anti-GAD65 and anti-IA2 autoantibodies were negative. As a particularity, the child presented episodes of psychomotor agitation. In his case, the family history was positive, the father’s brother and sister and both paternal grandparents being diagnosed with DM (not being genetically tested) (Figure 3).

The second patient (P02, C.I., aged 7 years and 8 months) was evaluated for the investigation of elevated fasting plasma glucose (maximum value of 130 mg/dL) associated with an HbA1c of 6.3% and 2 h glucose postprandial of 150 mg/dL, in the absence of autoimmunity elements (absent pancreatic islet autoantibodies). The family history was positive: the child’s mother had gestational diabetes mellitus (GDM) treated with insulin, and the child’s maternal grandfather was diagnosed with diabetes (considered type 2), treated with oral antidiabetics (OA) (not being genetically tested) (Figure 3). The child also had attention deficit hyperactivity disorder (ADHD), being treated with Atomoxetine (Strattera). WES testing at the Blueprint Genetics Laboratory identified the heterozygous pathogenic variant *GCK* c.106C>T, (p.Arg36Trp) (Table 2).

The third patient (P03, D.P.), currently aged 15 years and 6 months, was diagnosed with prediabetes in 2013 (at the age of 4) based on an average plasma glucose value of 150 mg/dL associated with HbA1c of 6.7% and a plasma C-peptide level of 0.51 ng/dL (Table 3).

Initially, a diet was recommended, and, later, basal insulin therapy was instituted. The family history was positive; the patient’s father and both grandparents also had DM (considered type 2) treated with OA (not having been genetically tested). Afterwards, the patient’s brother was diagnosed with type 1 diabetes mellitus (T1DM) with increased insulin requirements (Figure 3). During the attempt to gradually stop insulin therapy, a slight increase in blood glucose and HbA1c levels was observed, and, at the request of the parents, it was decided to maintain a minimal dose of basal insulin therapy. In addition, the child also presented sacral spina bifida which did not require surgical treatment. Genetic testing of the patient and his brother (Invitae Monogenic Diabetes Panel) identified the *GCK* c.106C>T variant in the patient, the mutation being absent in his brother’s case.

The heterozygous variant *GCK* c.106C>T, (p.Arg36Trp) is reported in the gnomAD database [50,51] in four heterozygous patients and is also found in the ClinVar database (ID 431973) [52]. The variant is predicted to be pathogenic by all in silico tools used. The *GCK* c.106C>T variant was first reported to be associated with autosomal dominant hyperglycemia in a French family in which the mutation occurred de novo (PMID: 8168652 [16]. Later, the variant was reported in several patients and families with MODY or mild hyperglycemia (PMID: 25555642, 35592779, 28012402, and 33477506) [17,18,19,20].

Fendler et al. identified this variant in 11 of 68 Polish patients with *GCK*-MODY (PMID: 21521320) [21]. Valentínová et al. reported this variant in a Slovak MODY family in which all affected family members had the heterozygous genotype (PMID: 22493702) [22]. In a functional study, the p.Arg36Trp mutant protein had similar kinetic properties to the wild-type protein and no effect on thermal stability compared to the wild-type (PMID: 10426385) [23].

The fourth patient (P04, S.L.), aged 9 years and 3 months, was evaluated due to the detection of a plasma glucose value of 200 mg/dL in the absence of other clinical symptoms. The clinical examination did not reveal any pathological elements, with signs of insulin resistance being absent (obesity, acanthosis nigricans, or metabolic syndrome). Anti-GAD65 and anti-IA2 antibodies were negative, and the plasma C-peptide level was normal. Molecular genetic testing (next-generation sequencing, NGS) using the Blueprint Genetics (BpG) MODY Panel identified the pathogenic variant *GCK* c.1072C>T (p.Arg358*) which is a nonsense mutation that predicts the substitution of an arginine amino acid for a premature stop codon at position 358 of the protein, affecting one functional domain. This sequence change creates a premature translational stop signal (p.Arg358*) in the *GCK* gene. It is expected to result in an absent or disrupted protein product. Loss-of-function variants (LOF) in the *GCK* gene are known to be pathogenic (PMID: 10754480, 14578306, 24323243, and25015100) [24,25,26,27].

It is described in the Human Gene Mutation Database (HGMD) database (CM002026) as a pathogenic variant associated with MODY type 2 [53]. The variant is described in the dbSNP database (rs780716926) [54] and it is also described in the population frequency databases gnomAD (0.00043%) [50]. The bioinformatic predictor MutationTaster estimates that the change would have a pathogenic effect. The variant found in the patient is reported in the scientific literature consulted in two cases in cohorts of the Spanish population with MODY (PMID: 10754480) (ESPE Abstracts (2019) 92 RFC1.4) [24]. The family history revealed that the patient’s father had occasional moderate hyperglycemia without any treatment.

The fifth patient (P05, A.L.) at the age of 3 years and 5 months presented at the onset specific symptoms of diabetes (polyuria, polydipsia, and weight loss) associated with high plasma glucose levels (average plasma glucose level of 148 mg/dL) and HbA1c of 6.8%. Plasma C-peptide had normal levels, and anti-GAD65 and anti-IA2 antibodies were negative, ruling out a form of type 1 diabetes mellitus (T1DM). The family history was positive, the mother being diagnosed with insulin-requiring gestational diabetes, and the maternal grandmother diagnosed with type 2 diabetes mellitus (T2DM). The genetic analysis (Invitae Monogenic Diabetes Panel) identified a heterozygous pathogenic variant in the *GCK* gene c.617C>T which determines the substitution of threonine with methionine at codon 206 of the GCK protein (p.Thr206Met). The threonine residue is highly conserved and there is a moderate physicochemical difference between threonine and methionine. At the time of genetic testing, this genetic variant was considered de novo, not being reported in the literature. It was later reported in international databases, being classified as pathogenic (ClinVar, Variation ID: 1191898; gnomAD; dbSNP database: rs1441649062) [50,52,54]. This variant was detected in patients with MODY with an autosomal dominant transmission in several studies (PMID: 34440516, 31216263, 28726111, 24606082, 19790256, and 16173921) [28,29,30,31,32,33].

Algorithms developed to predict the effect of missense changes on protein structure and function (SIFT, PolyPhen-2, Align-GVGD) suggested that this variant is likely to disrupt the function of encoded proteins. Experimental studies have shown that this variant affects the function of the GCK protein (PMID: 16173921) [31], these results causing this variant to be classified as pathogenic. As a particularity, the patient also had agenesis of the inferior vena cava (absence of the infracardiac segment) and the profile for hereditary thrombophilia was positive: heterozygous for the *MTHFR* A1298C, *PAI-1* 675 4G/5G, and *EPCR* (Endothelial Protein C Receptor) A2/A3 variants—associated with a predisposition toward deep venous thrombosis.

In the case of patient P06, G.B., currently at the age of 10, the onset was at the age of 8 when hyperglycemia (160 mg/dL) was detected, associated with an HbA1c of 6.3%, C-peptide with normal values, and anti-GAD65 and anti-ICA antibodies that were negative. The child also had obesity and secondary dyslipidemia. The family history was positive, both maternal and paternal grandparents being diagnosed with DM (not being genetically tested) (Figure 3). Genetic testing (NGS) using the Blueprint Genetics (BpG) MODY Panel identified a heterozygous splice donor variant *GCK* c.1019+1G>A (Table 3).

This variant is absent in gnomAD, a large reference population database (n > 120,000 exomes and >15,000 genomes) which aims to exclude individuals with severe pediatric disease [50]. This variant is reported in the ClinVar database (variation ID 2664360) [52]. This variant substitutes a nucleotide within a canonical splice site and is, therefore, likely to lead to abnormal splicing. The impact of this variant on the encoded protein cannot be verified without transcriptional studies. However, the variant is predicted to lead to the in-frame skipping of an exon, resulting in the loss of 52 amino acids. Pathogenic missense variants have been reported within this exon (PMID: 34826540, 36257325, 14517956, and 15928245), indicating that this region of the protein is functionally important [34,35,36,37]. This *GCK* c.1019+1G>A has been reported in multiple patients with GCK-related phenotypes (PMID: 15928245, 28012402, 32533152, 31638168, and 31529753) [20,30,38,39,40].

In addition, several other variants affecting the same splice donor site, c.1019+5G>C, c.1019+5G>A, c.1019+3dupG, c.1019+2T>G, c.1019+2T>C, c.1019+2T>A, and c.1019+1delG, have been reported in patients with GCK-related disease (PMID: 14578306, 19564454, 31638168, 29056535, 31638168, and 27908292,) [27,38,41,42,43,44].

Patient 07, F.L.Z., was diagnosed at the age of 15 years and 9 months when he presented with polyuria with nocturia, polydipsia, and elevated blood glucose levels. In addition, the patient was severely obese with a BMI of 37.4 kg/m^2^ (weight excess of 45–50 kg compared to the weight corresponding to the height). The diagnosis of DM was biologically confirmed (an HbA1c value of 11.5% was detected) and basal-bolus therapy (BBT) using an ultra-rapid insulin analogue and insulin glargine 300 U was initiated.

The plasma C-peptide level of 2.62 ng/dL was normal, and anti-GAD65 and anti-ICA antibodies were negative (Table 3). Considering the positive family history (the patient’s father was recently diagnosed with T2DM and, in the absence of signs of autoimmunity, the suspicion of MD was raised. The molecular testing (Invitae Monogenic Diabetes Panel) of family members revealed the presence of a heterozygous pathogenic variant in the *KCNJ11* c.616C>T (p.Arg206 Cys) in the patient and his mother (asymptomatic), while the patient’s father and sister did not present the respective mutation. The family history revealed that the maternal grandfather had also been diagnosed with insulin-requiring diabetes.

Taking into account the result of the genetic test, the treatment scheme was reconsidered, and insulin therapy was replaced with sulfonylureas (gliclazide) at a minimum dose. Two months after starting the treatment with oral antidiabetics (OA), the patient came for control when it was found that the obesity had become worse (weight = 137 kg, BMI = 38 kg/m^2^), along with showing borderline hypertension, and poor glycemic control with an HbA1c level of 11.2%. The doses of sulfonylureas were subsequently increased and basal insulin treatment (INN-insulin degludec) and a hypocaloric diet were associated.

The *KCNJ11* c.616C>T variant located in exon 1 of the gene is a missense mutation that determined the substitution of arginine, which is basic and polar, with cysteine, which is neutral and slightly polar, in codon 206 of the KCNJ11 protein (p.Arg206Cys). This variant is present in international databases (rs775204908, gnomAD 0.01%; ClinVar ID: 1338615) [50,52,54] and has been reported in several studies in individuals with autosomal dominant familial hyperinsulinism (PMID: 25555642, 30026763, 12524280, 31464105, and 23348805) [19,45,46,47,48].

Algorithms developed to predict the effect of missense changes on protein structure and function (SIFT, PolyPhen-2, Align-GVGD) suggested that this variant probably disrupts protein function, and experimental studies confirmed that the respective mutation disrupts the amino acid residue p.Arg206 from the protein Kir6.2 subunit of the K_ATP_ channel in the pancreatic β-cells (PMID: 12524280) [46].

Other variants that disrupt this residue have been determined to be pathogenic (PMID: 31464105; Invitae) [47]. This suggests that this residue is clinically significant and that variants that disrupt this residue are likely to cause disease. For these reasons, this variant was classified as pathogenic.

Considering that the patient’s mother had the same mutation but did not show symptoms specific to diabetes, being asymptomatic, the variable expressivity and incomplete penetrance of the mutation identified in her was discussed.

The last patient (P08, E.T.), female, aged 12 years, was investigated following the detection of elevated plasma glucose values up to 189 mg/dL associated with glycosuria. Clinically, the patient is also obese (BMI of 24.6 kg/m^2^, 97th percentile). Biological investigations revealed a plasma level of HbA1c of 7.4%, the plasma C-peptide level was normal, and anti-GAD65 and anti-ICA antibodies were absent. Although the family history was negative for diabetes, the genetic testing of the patient was still recommended (Invitae Monogenic Diabetes Panel). Thus, a heterozygous pathogenic variant was identified in exon 7 of the *HNF4A* gene c.733C>T (p.Arg245Cys), present in the ClinVar database (Variation ID: 804917) [52]. This missense mutation caused the substitution of arginine, which is basic and polar, with cysteine, which is neutral and slightly polar, in codon 245 of the HNF4A protein (p.Arg245Cys).

The frequency of this variant in the general population cannot be estimated due to insufficient data related to this variant in the gnomAD database [50]. The *HNF4A* c.733C>T variant has been reported in individuals with both familial hyperinsulinism and/or MODY (PMID: 30026763, 23348805, and 31957151; Invitae) [45,48,49]. In at least one patient, the mutation was de novo, the family history being negative.

This aspect suggests that we can suspect a diagnosis of MODY even in the absence of a positive family history for diabetes in patients who do not present autoimmune markers for T1DM.

Algorithms developed to predict the effect of missense changes on protein structure and function (SIFT, PolyPhen-2, Align-GVGD) suggested that this variant is likely to be pathogenic.

Taking into account that the patient’s parents were asymptomatic, with genetic test being negative, the patient most likely has a de novo mutation, not being able to exclude germline mosaicism in one of the parents.

The initial treatment involved a diet regimen with sulfonylureas (glicazide). Because at the minimum dose of OA, the patient had episodes of hypoglycemia, it was decided to stop the drug treatment. Under the diet, the plasma level of HbA1c was maintained at values of 5.8%. It was not possible to carry out the genetic testing of all family members of the index patients, especially those diagnosed with diabetes. In only two of the families (P07 and P08) was genetic testing possible for all first-degree relatives, and, partially, in the case of patient P03, where only his brother was tested, not his parents. We believe that genetic testing is necessary, especially in families with *GCK*-MODY 2 in which the treatment could be revised, giving up the antidiabetic medication.

We consider that the small number of analyzed cases is a limitation of our study; our results cannot be extended to all patients with a suspicion of MODY, coming from the region of Moldova. This aspect is caused by the lack of molecular testing possibilities for all patients meeting the clinical criteria for MODY, including their family members (the cost of genetic analysis being borne by the family). Added to this are the difficulties encountered in obtaining the most accurate information related to the family history of diabetes, in some of the cases.

However, our results are consistent with those of other European studies, which included large cohorts of patients with MODY. It is known that the prevalence of MODY varies among different populations, and the genetic variants present in different populations may vary, depending on ethnicity.

Thus, Passanisi et al. [55] identified 37 patients (6.5%) with MODY from a number of 565 children and adolescents with diabetes (genetic tests for monogenic diabetes were performed in 68 patients). *GCK* MODY 2 was identified in 30 of the patients (81%), *HNF1A* MODY 3 in 5 patients (13.5%), and *HNF4A* MODY 1 in 2 of the patients (5.4%). The age of the patients varied between 0 and 16 years, with an average of 9.1 years [55].

Avelos et al. [56] identified 23 (50%) patients with MODY in 46 analyzed Portuguese families. *GCK* gene variants were present in 12 of the patients, *HNF1A* gene mutations in 8 patients, and *HNF4A* variants in 3 patients. Of these, sixteen were missense mutations, two nonsense, one frameshift, and one synonymous variant [56].

In a study of the Qatari population that included 37 patients meeting clinical criteria for MODY, ten missense mutations were identified in 24 patients. Most of the variants detected were located in *HNF1A* gene, the most frequent being rs587778397 (p.Arg177Trp). Variants of the *BLK* gene (rs766934515) were identified in five patients, and mutations of the *GCK* gene were present in four of the patients. Other rare mutations were located in the *HNF4A*, *GCK*, *KLF11*, and *ABCC8* genes [57].

Ben Kelifa et al. [58] analyzed 23 unrelated Tunisian patients who presented with the clinical criteria for MODY. Patients were genetically tested (sequencing and MLPA) for mutations in the *GCK*, *HNF1A*, *HNF4A*, and *INS* genes. In three of the index cases (13.05%), mutations in the *GCK* and *HNF4A* genes were detected, while no patient presented mutations in the HNF1A and INS genes. The already known *HNF4A* c.-169C>T variant was detected in only one family (in the proband and his mother), while, in two unrelated subjects, a new *HNF4A* c.-457C>T mutation was identified. The segregation of this mutation in the two families was not possible because its members refused genetic testing [58]. In two other unrelated patients, a new *GCK* c.457C>T mutation was identified. The authors’ conclusion was that, despite the restrictive clinical criteria used to select patients, mutations in the most common genes known to be associated with MODY do not explain the majority of cases in Tunisians. This suggests that, in the Tunisian population, there could be other unidentified candidates that contribute to the etiology of MODY [58].

Bario et al. [59] identified 14 mutations in three genes associated with MODY in 22 pediatric patients of Spanish origin. Nine of these were new mutations and all of them cosegregate with the clinical phenotype of MODY within the pedigrees. *GKG* MODY 2 mutations were the most frequent (41% of cases): seven of these were de novo mutations (R369P, S411F, M298K, C252Y, Y108C, A188E, and S383L) and two already known mutations. In four cases (18%), variants in *HNF1A* MODY 3 were identified (including the new mutation R27G). The *HNF4A* IVS5-2delA (MODY 1) variant was identified in a single family (4%), being reported for the first time in the Spanish population [59].

The new challenges and future perspectives in MODY are as follows:

From a clinical point of view, increasing the accuracy of diagnostic methods will allow us to distinguish MODY from other forms of diabetes (T1DM and T2DM), thus avoiding unnecessary treatment with insulin or sulfonylurea that can affect the patient’s health. In the future, new research and studies on translational biology and integrative genomics will be needed for monogenic and polygenic forms of diabetes, which will provide new perspectives on the complex molecular mechanisms involved in the pathophysiology, as well as regarding the treatment of diabetes [60].

The use of NGS in the diagnosis of MODY represents a key element in the identification of the etiological factors of MODY, allowing the identification of both already known genes or new candidate genes, which are the basis of pancreatic β-cell dysfunction. Thus, it would be possible to elucidate the etiology of some forms of familial or atypical early-onset diabetes [60].

A transcriptomic analysis can provide significant information related to the molecular mechanisms underlying the appearance of MODY. Another direction to approach in studying and differentiating MODY from other types of diabetes could be metabolomic profiling, while the use of pluripotent stem cells (PSCs) could be useful in deciphering the molecular mechanisms underlying different forms of diabetes, including MODY [60].

### Genetic Counseling in Patients with MODY

Genetic counseling plays a major role in the management of patients with MODY. In the analyzed group, the family history was positive for diabetes mellitus in seven of the eight patients studied. In their case, the risk of recurrence in the first-degree relatives of the patients is 50%, based on the autosomal dominant inheritance of the disease [61,62]. In the case of patient E.T. (P08), the family history was negative. Since the genetic variant present in the child was not identified in the parents (clinically healthy), most likely, a de novo mutation of the *HNF4A* gene occurred in him, without being able to exclude the presence of germinal mosaicism in one of the parents. Genetic counseling in the case of MODY patients and their families must respect bioethical principles. Most patients with MODY are children or young adults, and concern for the long-term well-being of both the patient and his family is essential. Information related to the diagnosis of the disease (including its confirmation by genetic testing) and its evolution, as well as the risk of recurrence in other family members or in the descendants of patients with MODY, must be communicated to patients and families in a clear, appropriate, non-directive manner [62]. Parents may feel discomfort when the genetic testing of their child is recommended, or the disclosure of information related to the child’s disease, including the result of the genetic test, to other family members who may also be at increased risk for MODY [63,64,65].

Genetic testing in family members of affected individuals allows the pre-symptomatic diagnosis of mutation carriers, for which preventive measures are required, including regular blood glucose monitoring, early diagnosis, and appropriate treatment [61,62,63].

## 4. Material and Methods

We analyzed a group of eight unrelated patients (including their families) with clinical suspicion of MODY, in the records of the Diabetes and Nutritional Diseases Clinic, from the Children’s Emergency Clinical Hospital, St. Maria Iași, Romania. All patients came from the same geographical region (Moldova area). The diagnosis was confirmed by extended genetic testing—targeted gene panel for monogenic diabetes (Blueprint Genetics (BpG), Hypoglycemia, Hyperinsulinism, and Ketone Metabolism Panel, respectively, and Invitae Hypoglycemia Panel) using next-generation sequencing (NGS) (Illumina Technology) or whole-exome sequencing (WES). The genetic analyses were carried out at laboratories abroad (Invitae and Blueprint Genetics); in Romania, it was not possible, at the time, to be tested in state hospitals through the national program of rare diseases.

The total genomic DNA was extracted from the biological sample using a bead-based method. Quantity of DNA was assessed using the fluorometric method. After assessment of DNA quantity, qualified genomic DNA sample was randomly fragmented using non-contact, isothermal sonochemistry processing. Sequencing library was prepared by ligating sequencing adapters to both ends of DNA fragments. Sequencing libraries were size-selected with a bead-based method to ensure optimal template size and amplified by polymerase chain reaction (PCR). Regions of interest (exons and intronic targets) were targeted using the hybridization-based target capture method. The quality of the completed sequencing library was controlled by ensuring the correct template size and quantity and to eliminate the presence of leftover primers and adapter–adapter dimers. Ready sequencing libraries that passed the quality control were sequenced using Illumina’s sequencing-by-synthesis method using paired-end sequencing (150 by 150 bases). Primary data analysis converting images into base calls and associated quality scores was carried out by the sequencing instrument using Illumina’s proprietary software, generating CBCL files as the final output. Base-called raw sequencing data were transformed into FASTQ format using Illumina’s software (bcl2fastq). Sequence reads of each sample were mapped to the human reference genome (GRCh37/hg19). Burrows–Wheeler Aligner (BWA-MEM) software was used for read alignment. Duplicate read marking, local realignment around indels, base quality score recalibration, and variant calling were performed using GATK algorithms (Sentieon) for nDNA. Variant data were annotated using a collection of tools (VcfAnno and VEP) with a variety of public variant databases, including but not limited to gnomAD, ClinVar, and HGMD. The patient’s sample was subjected to thorough quality control measures including assessments for contamination and sample mix-up. Copy number variations (CNVs), defined as single-exon or larger deletions or duplications (Del/Dups), were detected from the sequence analysis data using a proprietary bioinformatics pipeline. CNVs (Deletions/Duplications) were confirmed using a digital PCR assay if they covered less than 10 exons (heterozygous).

The pathogenicity of the identified gene variants was assessed according to the American College of Medical Genetics and Genomics and Association for Molecular Pathology (ACMG/AMP) guidelines. For the interpretation of the variants identified in patients with MODY, we also used the HGMD Professional and ClinVar databases.

The potential pathogenicity of novel missense variants was determined using three in silico prediction methods: PolyPhen-2 (http://genetics.bwh.harvard.edu/pph2/), PROVEAN (http://provean.jcvi.org/genome_submit_2.php?species?human), and MutationTaster (http://www.mutationtaster.org/ChrPos.html). In case of identification of CNVs, the Database of Genomic Variants and DECIPHER were analyzed, as well as specialized literature to evaluate their clinical relevance. 

After the molecular classification of the mutations, for each patient and his family, we analyzed the data obtained through the family anamnesis, including the genealogical tree, as well as the correlation between the metabolic phenotype (clinical and paraclinical) and the type of mutation detected. We also compared the results obtained with the data from the literature, highlighting, at the same time, the particularities of each patient in the context of his family. The study was conducted according to the Helsinki II Declaration and it was approved by the Ethics Committee of the Children’s Emergency Clinical Hospital, St. Maria Iași, Romania (Certificate no. 14367/2024).

In the case of each child, the informed consent of both parents was obtained, as well as of all adults who were clinically evaluated and who underwent genetic testing.

## 5. Conclusions

In conclusion, we used the targeted gene sequencing of some gene panels, and, in some cases, WES, with the aim of identifying the involved gene variants (already known or new gene variants) in the case of a group of patients in whom there was a clinical suspicion of MD/MODY.

*GKC*-MODY 2 was the most frequently detected variant, but rare forms of *KCNJ11*-MODY 13, specifically, *HNF4A*-MODY 1, were also identified. Our study highlights the fact that the use of high-performance molecular technology (NGS) and gene panels or WES in the diagnosis of monogenic diabetes is an essential factor for the correct identification of MODY subtypes, both in patients and in other family members. In the absence of genetic testing, the diagnosis could have been delayed, taking into account the young age of onset of the disease in most of the patients included in the study, as well as due to the absence of associated clinical symptoms, and, in some cases, a negative family history. The application of precision medicine, including genetic tests in the diagnosis of monogenic diabetes, including MODY, must represent a standard method in the management of patients with this pathology, especially since there is a correlation between the genetic variants, the patients’ phenotype, and the treatment used.

Identifying genetic biomarkers for monogenic diabetes correlated with race and ethnicity will be a challenge for future research.

Expanding genetic testing to larger and more diverse populations and performing functional studies of genetic variants, together with establishing robust ethical frameworks and counseling protocols for genetic testing in MODY, could contribute to early diagnosis and personalized treatment correlated with genetic etiology. The implementation of a guideline in the case of patients with MODY could facilitate genetic testing. Moreover, future research should explore best practices for genetic counseling, especially in cases where there is an increased risk for MODY in other family members. Genetic testing would allow pre-symptomatic diagnosis in the case of carriers of genetic variants associated with MODY, in which preventive measures are required, that include monitoring the plasma glucose level and diet.

## Figures and Tables

**Figure 1 ijms-25-06318-f001:**
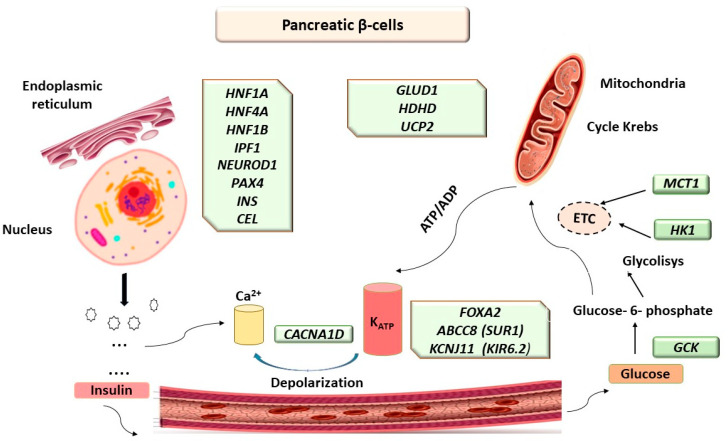
Molecular mechanism and genetic heterogeneity in monogenic diabetes (MD) and maturity-onset diabetes of the young (MODY).

**Figure 2 ijms-25-06318-f002:**
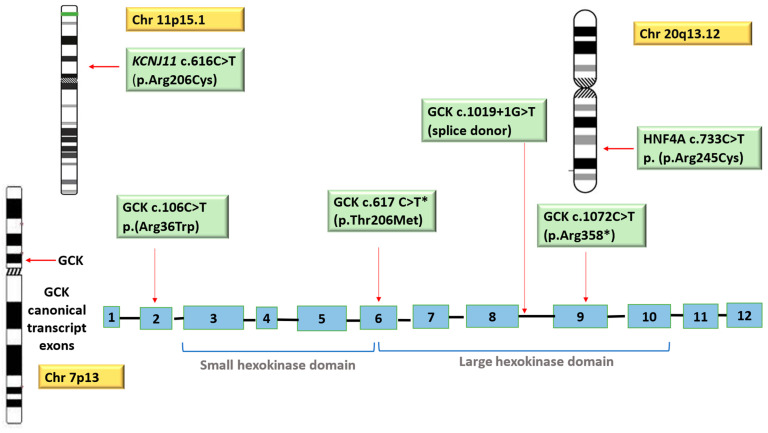
The spectrum of genetic variants detected in patients with MODY.

**Figure 3 ijms-25-06318-f003:**
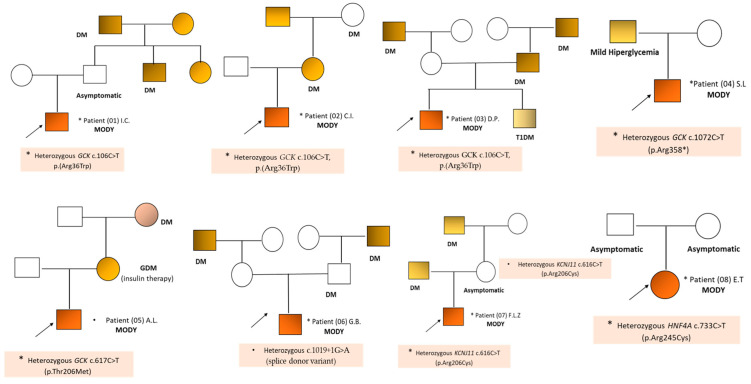
Family trees and data collected in the case of patients with MODY. DM: Diabetes mellitus (not genetically tested); T1DM: Type 1 diabetes mellitus; MODY: Maturity-onset diabetes of the young; GDM: Gestational diabetes mellitus.

**Table 1 ijms-25-06318-t001:** Genetic heterogeneity correlated with variable phenotype in MODY [1,3,6,8,9,10,11].

MODY Type	Gene Function	OMIM	Locus	Pathophysiology	Other Features	Treatment
*HNF4A*-MODY 1 (5–10%)	TF	600281	20q13.12	β-CD	Hyperinsulinism during infancy, low TG level	Sulfonylureas
*GCK*-MODY 2 (30–50%)	Glucokinase (Hexokinase 4)	138079	7p13	β-CD	Mild fasting hyperglycemia	Diet
*HNF1A*-MODY 3 (30–65%)	TF	142410	12q24.31	β-CD	Glycosuria	Sulfonylureas
*PDX1*-MODY 4 (1%)	TF	600733	13q12.2	β-CD	Pancreatic agenesis in homozygote/compound heterozygote	Diet or OAD or insulin
*HNF1B*-MODY 5 (<5%)	TF	189907	17q12	β-CD	Renal/genital anomalies, pancreatic hypoplasia	Insulin
*NEUROD1*-MODY 6 (<1%)	TF	601724	2q31.3	β-CD	Susceptibility for T2DM in heterozygous; neonatal diabetes; neurological abnormalities in homozygous mice	OAD or insulin
*KLF11*-MODY 7 (<1%)	TF	603301	2p25.1	β-CD	Early-onset T2DM	OAD or insulin
*CEL*-MODY 8 (<1%)	Controls pancreatic functions; plays essential role in digestion and intestinal absorption of dietary lipids	114840	9q34.13	Pancreas endocrine and exocrine dysfunction	Lipomatosis	OAD or insulin
*PAX4*-MODY 9 (<1%)	TF	167413	7q32.1	β-CD	Ketoacidosis	OAD, diet, or insulin
*INS*-MODY 10 (<1%)	Encode the proinsulin precursor	176730	11p15.5	Severe insulin deficiency	PND	OAD, diet, or insulin
*BLK*-MODY 11 (<1%)	Encodes B-lymphoid TK	191305	8p23.1	Abnormal IS	Overweight	OAD, diet, or insulin
*ABCC8*-MODY 12 (<1%)	Regulation of IS	600509	11p15.1	K_ATP_ dysfunction	PND, TND, HHF1	Sulfonylurea
*KCNJ11*-MODY 13 (<1%)	Regulation of IS	600937	11p15.1	K_ATP_ dysfunction	HHF2, PND2, TND3	OAD or insulin
*APPL1*-MODY 14 (<1%)	Insulin signaling pathways	604299	3p14.3	Insulin IS defect	Higher adiponectin levels among T2DM patients	Diet, OAD, or inulin

MODY: Maturity-onset diabetes of the young; OAD: Oral antidiabetic agents; PND: Permanent neonatal diabetes; TND: Transient neonatal diabetes; T2DM: Type 2 diabetes mellitus; HHF1: Hyperinsulinemic hypoglycemia, familial, 1; *HNF4A:* Hepatocyte nuclear factor 4 alpha; *GK*: Glucokinase; *HNF1A*: Hepatocyte nuclear factor 1 alpha; TF: Transcription factor; β-CD: β-cell dysfunction; *PDX1*: Pancreatic and duodenal homeobox 1; *HNF1B:* Hepatic nuclear factor 1 beta: *NEUROD1:* Neurogenic differentiation 1; *KLF11*: Kruppel-like factor 11; *CEL*: Carboxyl-ester lipase; *PAX4*: Paired box gene 4; *INS*: Insulin; *BLK:* BLK protooncogene, SRC family tyrosine kinase; *ABCC8:* ATP-binding cassette, subfamily C, member 8; K_ATP_: Adenosine triphosphate (ATP)-sensitive potassium channel; *KCNJ11:* Potassium channel, inwardly rectifying, subfamily J, member 11; HHF2: Familial hyperinsulinemic hypoglycemia 2; *APPL1:* Adaptor protein, phosphotyrosine interaction, Ph domain, and leucine zipper-containing protein 1; IS: Insulin secretion; TK: Tyrosine kinase; TG: Triglyceride.

**Table 2 ijms-25-06318-t002:** Genetic heterogeneity and genotype–phenotype correlations in patients with MODY.

Patient ID	Mutation	Exon	Genotype	Effect of Mutation/ Pathogenicity	Protein	GnomAD	dbSNP Database	ClinVar	Previous Studies
P01 (I.C)	*GCK* c.106C>T	exon 2	Hz	Missense/Pathogen	p.Arg36Trp	gnomAD (0.008%)	rs762263694	Variation ID 431973	[16,17,18,19,20,21,22,23]
P02 (C.I)	*GCK* c.106C>T	exon 2	Hz	Missense/Pathogen	p.Arg36Trp	gnomAD (0.008%)	rs762263694	Variation ID:431973	[16,17,18,19,20,21,22,23]
P03 (D.P)	*GCK* c.106C>T	exon 2	Hz	Missense/Pathogen	pp.Arg36Trp	gnomAD (0.008%)	rs762263694	Variation ID:431973	[16,17,18,19,20,21,22,23]
P04 (S.L.)	*GCK* c.1072C>T	exon 9	Hz	Nonsense/Pathogen	p.Arg358*	gnomAD (0.00043%)	rs780716926		[24,25,26,27]
P05 (I.L)	*GCK* c.617C>T	exon 6	Hz	Missense/Pathogen	p.Thr206Met	gnomAD (0.000004)	rs1441649062	Variation ID:1191898	[28,29,30,31,32,33]
P06 (G.B)	*GCK* c.1019+1G>A	Intron 8 (splice donor)	Hz	Splice_donor_variant		gnomAD (0/0)		Variation ID 2664360	[34,35,36,37,38,39,40,41,42,43,44]
P07 (F.L.Z)	*KCNJ11* c.616C>T	exon 1	Hz	Missense/Pathogen	p.Arg206Cys	gnomAD (0.01%)	rs775204908	Variation ID:1338615	[45,46,47,48]
P08 (E.T)	*HNF4A* c.733C>T	exon 7	Hz	Missense/Pathogen	p.Arg245Cys	gnomAD (0.00001)	rs1290868034	Variation ID: 804917	[46,49,50]

Hz: Heterozygous genotype.

**Table 3 ijms-25-06318-t003:** Clinical and paraclinical data of patients with MODY.

Criteria	P01 (C.I)	P02 (C.I.)	P03 (D.P.)	P04 (S.L)	P05 (I.L.)	P06 (G.B)	P07 (F.L.Z)	P08 (E.T.)
**Age of presentation (years)**	2.5 y	7.6 y	4y	9.2 y	3.4 y	10 y	15.9 y	12 y
**Follow-up** **months (m)/years (y)**	7 m	3 m	11 y 4 m	-	3 m	2 y	2 y	2 y 3 m
**Sex**	M	M	M	M	M	M	M	F
**Clinical symptoms**	-	-	-	-	+	+	+ (DKA)	-
**Higher hyperglycemia level (mg/dL)**	150	150	140	200	148	160	248	189
**Obesity**	-	-	-	-	-	+	+	+
**Initial HbA1c**	5.9%	6.3%	6.2%	6.5%	6.8%	6.3%	11.5%	7.4%
**Type of autoantibodies tested/results**								
GAD65	-	-	-	-	-	-	-	-
ICA	-	-	-	-	-	-	-	-
IA-2	-				-	-		
**C-peptide (ng/mL)**	normal	normal	low	normal	normal	normal	normal	normal
**Family history of diabetes**	+	+	+	+	+	+	+	-
**Pharmacological Treatment**	-	-	+	-	-	-	+	+
**Type of treatment**	diet	diet	insulin	diet	diet	diet	insulin + sulfonylurea	sulfonylurea
**Last HbA1c**	6%	6%	6.7%	6.1%	6.8%	6.3%	11.2%	5.8%

P: Patient; M: Male; F: Female; DKA: Diabetic ketoacidosis; HbA1c: Glycosylated hemoglobin; GAD65: The glutamic acid decarboxylase 65-kilodalton isoform (GAD65) antibody; ICA antibodies: Islet cell antibodies; IA-2 antibodies: Tyrosine phosphatase-like protein IA-2 (IA-2A) antibodies.

## Data Availability

The data are contained within the article.

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
