# Peer review of "The Importance of Molecular Genetic Testing for Precision Diagnostics, Management, and Genetic Counseling in MODY Patients"

_ijms, 2024, doi:10.3390/ijms25126318_

Round 1

Reviewer 1 Report

Comments and Suggestions for Authors

When you refer to  "modern technologies" do you mean Next Generation Sequencing technologies''?. Please, specify.

In table 1 could the authors indicate the % relative to total mutation cases (estimated according to the literature).

The study is based on only 8 patients, that are not representative of the full MODY cases in the population. This is a limitation.

All the patients have a family history of diabetes, but there are no confirmation of the likely pathogenic variants in the affected family members.

Why there are no methods section, reporting (briefly) the sequencing technique you have used?. illumina, ion-torrent, other?.

Did the authors sequenced a larger cohort and found cases with variants of uncertain significance or no variant identified?. If so i think the ms could be improving by including these patients, at least indicating the numbers.

Comments on the Quality of English Language

revise for typos. Minor revision.

Author Response

Response to Reviewer 1 Comments

Thank you very much for taking the time to review this manuscript. Please find the detailed responses below and the corresponding revisions/corrections highlighted (red color) in the re-submitted files.

  1. When you refer to "modern technologies" do you mean Next Generation Sequencing technologies''?. Please, specify.

R: Yes. We referred to next-generation sequencing (NGS). I have completed the explanation in the text (marked in red).

  1. In table 1 could the authors indicate the % relative to total mutation cases (estimated according to the literature).

R: Thank you for your suggestion. I have added the percentages in the table for each type of MODY.

  1. The study is based on only 8 patients, that are not representative of the full MODY cases in the population. This is a limitation. All the patients have a family history of diabetes, but there are no confirmation of the likely pathogenic variants in the affected family members.

R: Thank you for your observation. Indeed, the small number of cases represents a limitation of our study. I added / explained this in the article (marked in red in the text).

  1. Why there are no methods section, reporting (briefly) the sequencing technique you have used?. illumina, ion-torrent, other?. Did the authors sequenced a larger cohort and found cases with variants of uncertain significance or no variant identified?. If so i think the ms could be improving by including these patients, at least indicating the numbers.

R: Our study included only patients who met the clinical criteria of MODY and in whom the diagnosis of MODY could be genetically confirmed (since the parents could afford to pay the cost of the analysis – gene panels or WES). I added the explanation in the text related to the fact that unequal access to genetic tests delays the diagnosis or leads to the wrong diagnosis.

Since the genetic testing was done at laboratories abroad (Invitae and Blueprint genetics), we do not have detailed data related to the laboratory methods used other than those already mentioned in the article. Illumina technology was used for sequencing (see note in the material and method section).

Please check the new version of the article that I have attached. Thank you once again for the effort you put into revising our article.

Dr. Butnariu

Reviewer 2 Report

Comments and Suggestions for Authors

The manuscript, “The Importance of Molecular Genetic Testing for Precision Diagnostics, Management, and Genetic Counseling in MODY Patients,” investigates the role of genetic testing in diagnosing and managing Maturity-Onset Diabetes of the Young (MODY). The study highlights the importance of identifying genetic variants associated with MODY to ensure accurate diagnosis, personalized treatment, and appropriate genetic counseling. The research is well-structured, and the findings are significant for advancing the understanding and clinical management of MODY.

Overall, the manuscript is well put together and provides a detailed background on MODY, discussing its classification, genetic basis, and clinical significance. The authors have effectively summarized the current knowledge. The study employs robust genetic testing methods, including next-generation sequencing and whole exome sequencing, to identify genetic variants in patients suspected of having MODY. The use of these advanced genomic techniques ensures high accuracy and reliability of the findings. The paper further underscores the clinical implications of genetic testing in MODY patients, such as early diagnosis, subtype differentiation, and personalized treatment plans. The emphasis on genotype-phenotype correlations and family context adds depth to the clinical utility of the study.

While the study is limited by the small sample size from geographical and ethnic homogeneous populations (I presume this, but these details need to be provided), this small sample size does not take away from the overall findings/impact of the study. Additionally, the study’s clinical data seems to be based on self-reported family histories and medical records, possibly subject to recall bias and inaccuracies. Prospective studies with more rigorous clinical data collection methods would be ideal here.

The results section is well-organized, presenting comprehensive data on the identified genetic variants and their clinical manifestations. The inclusion of tables and figures aids in the clear presentation of complex genetic information. The identification of GCK-MODY 2 as the most frequent variant and the detection of rare forms such as KCNJ11-MODY 13 and HNF4A-MODY 1 are significant findings of the study. Overall, the study’s findings highlight the genetic heterogeneity of MODY and the need for precision diagnostics.

While the manuscript presents a thorough investigation into the genetic basis and clinical management of MODY, some areas could be enhanced to strengthen its overall impact. Expanding the introduction to further emphasize the diagnostic challenges and limitations associated with MODY would provide a stronger rationale for the study. Lines 20-23 and 51 -53 repeat: “The diagnosis of MODY still remains a challenge for clinicians, with many cases being misdiagnosed as type 1 or type 2 diabetes mellitus (T1DM / T2DM), and over 80% of cases remain undiagnosed.” This could be rephrased for clarity. Further, in the introduction, the authors should consider specifying why these misdiagnoses occur and the implications for patient care.

The discussion section would benefit from a deeper exploration of the findings within a broader clinical context, including comparisons with other studies or populations to elucidate the genetic diversity of MODY. Incorporating a brief discussion on the ethical considerations of genetic testing and counseling, particularly regarding disclosing genetic risk to family members, would add valuable context. Additionally, the conclusion should be expanded to suggest future research directions and study limitations. Perhaps suggestions on expanding genetic testing to larger and more diverse populations, conducting functional studies on genetic variants, establishing robust ethical frameworks, and counseling protocols for genetic testing in MODY could be suggested here. Future research should explore the best practices for genetic counseling, particularly in cases where genetic risk may impact family members.

Minor language and formatting adjustments must be made throughout the manuscript to enhance clarity and readability. Tables 1 and 2 are highly informative but could benefit from more descriptive titles and captions. Figure captions should be descriptive enough to help readers understand the figure independently of the main text. This would help readers quickly grasp the content and significance of the tables and figures.

Overall, this manuscript presents valuable insights into the genetic basis of MODY and underscores the importance of molecular genetic testing in its diagnosis and management. With revisions, this paper has the potential to significantly contribute to the fields of endocrinology and genetic counseling. I recommend its publication after addressing the above suggestions.

Comments on the Quality of English Language

Overall, the English language is of good quality. Minor language and formatting adjustments must be made throughout the manuscript to enhance clarity and readability, as suggested in the comments to the authors. 

Author Response

Response to Reviewer 2 Comments

Thank you very much for taking the time to review this manuscript. Please find the detailed responses below and the corresponding revisions/corrections highlighted (red color) in the re-submitted files.

  1. While the manuscript presents a thorough investigation into the genetic basis and clinical management of MODY, some areas could be enhanced to strengthen its overall impact. Expanding the introduction to further emphasize the diagnostic challenges and limitations associated with MODY would provide a stronger rationale for the study. Lines 20-23 and 51-53 repeat: “The diagnosis of MODY still remains a challenge for clinicians, with many cases being misdiagnosed as type 1 or type 2 diabetes mellitus (T1DM / T2DM), and over 80% of cases remain undiagnosed.” This could be rephrased for clarity. Further, in the introduction, the authors should consider specifying why these misdiagnoses occur and the implications for patient care.

R: Thank you for your recommendation. I added the explanation in the article (marked in red color in the text).

  1. The discussion section would benefit from a deeper exploration of the findings within a broader clinical context, including comparisons with other studies or populations to elucidate the genetic diversity of MODY.

R: Thank you for your recommendation. Although our study is limited by the small number of cases presented, at your suggestion we have compared some studies from the literature.

  1. Incorporating a brief discussion on the ethical considerations of genetic testing and counseling, particularly regarding disclosing genetic risk to family members, would add valuable context.

R: Thank you for your recommendation. I have added some information regarding the difficulties in giving genetic advice in MODY and the ethical problems that genetic testing in MODY raises, both for patients and other family members

4.Additionally, the conclusion should be expanded to suggest future research directions and study limitations. Perhaps suggestions on expanding genetic testing to larger and more diverse populations, conducting functional studies on genetic variants, establishing robust ethical frameworks, and counseling protocols for genetic testing in MODY could be suggested here. Future research should explore the best practices for genetic counseling, particularly in cases where genetic risk may impact family members.

R: Thank you for your recommendation. I added some information related to the challenges and future perspectives in MODY (in the discussion and conclusions sections) (marked in red in the text).

 Please check the new version of the article that I have attached. Thank you once again for the effort you put into revising our article.

Dr. Butnariu

Round 2

Reviewer 1 Report

Comments and Suggestions for Authors

All the patients they sent for genetic testing were mutation carriers?. There were no patients with negative results?. I assume that the inclusion criteria were highly strict thus limiting the rate of negatives, but most studies failed to identify pathogenic variants in 100% of the cases.

Apart of this, the authors aswered my questions.

Comments on the Quality of English Language

needs some editing, revision for typos.

Author Response

Thank you once again for the effort put into revising our article.
Indeed, our study was very restrictive, as it only included patients who met the clinical criteria for MODY and whose parents could afford to pay for genetic testing.
In our case, all genetically tested patients proved to be carriers of pathogenic mutations (100% detection rate).
We believe that if genetic testing had been possible in other patients with clinical suspicion of MODY, we would probably have detected cases without mutation or VUS.

Reviewer 2 Report

Comments and Suggestions for Authors

I am satisfied with the revisions incorporated by the authors and recommend the publication of the article.

Author Response

Thank you once again for the effort you put into revising our article.